# Prolonged Sleep Latency and Reduced REM Latency Are Associated with Depressive Symptoms in a Japanese Working Population

**DOI:** 10.3390/ijerph19042112

**Published:** 2022-02-13

**Authors:** Chie Omichi, Hiroshi Kadotani, Yukiyoshi Sumi, Ayaka Ubara, Kohei Nishikawa, Arichika Matsuda, Yuji Ozeki

**Affiliations:** 1Department of Psychiatry, Shiga University of Medical Science, Seta Tsukinowa-cho, Otsu City 520-2192, Japan; chie.omichi@ompu.ac.jp (C.O.); ysumi@belle.shiga-med.ac.jp (Y.S.); cykc1005@mail2.doshisha.ac.jp (A.U.); shiga@cbtcenter.jp (K.N.); arichika@belle.shiga-med.ac.jp (A.M.); ozeki@belle.shiga-med.ac.jp (Y.O.); 2Department of Epidemiology for Community Health and Medicine, Kyoto Prefectural University of Medicine, 465 Kajii-cho, Kamigyo-ku, Kyoto 602-8566, Japan; 3Department of Hygiene and Public Health, Osaka Medical and Pharmaceutical University, 2-7, Daigaku-machi, Takatsuki 569-8686, Japan; 4Graduate School of Psychology, Doshisha University, Kyoto 610-0394, Japan; 5Japan Society for the Promotion of Science, Research Fellowships, Tokyo 102-0083, Japan; 6Japan CBT Center, CG Building F4, 3-12 Chuo-cho, Hikone 522-0063, Japan

**Keywords:** electroencephalography, depression, REM, sleep latency, REM latency

## Abstract

Background: Examining the relationship between sleep and depression may be important for understanding the aetiology of affective disorders. Most studies that use electroencephalography (EEG) to objectively assess sleep have been conducted using polysomnography in the laboratory. Impaired sleep continuity, including prolonged sleep latency and changes in rapid eye movement (REM) sleep, have been reported to be associated with depression in clinical settings. Here, we aimed to use home EEG to analyse the association between sleep and depressive symptoms. Methods: We performed a cross-sectional epidemiological study in a large Japanese working population to identify the EEG parameters associated with depressive symptoms based on the results of a questionnaire survey and home EEG measurements using 1-channel (1-Ch) EEG. Results: The study included 650 Japanese patients (41.2% male, 44.7 ± 11.5 years) who underwent home EEG monitoring along with the Patient Health Questionnaire-9 (PHQ-9) to assess depressive symptoms. Logistic regression analysis revealed that depressive symptoms (PHQ-9 ≥ 10) were associated with sleep latency (odds ratio (OR) 1.02; 95% confidence interval (CI): 1.00–1.04) and REM latency (OR, 0.99; 95% CI: 0.99–1.00). Conclusions: Our results suggest that depressive symptoms are associated with prolonged sleep latency and reduced REM latency in a Japanese working population. The 1-Ch EEG may be a useful tool to monitor sleep and screen depression/depressive symptoms in non-clinical settings.

## 1. Introduction

Mental disorders are highly prevalent and contribute substantially to the total disease burden of the general population [1]. Patients with mental disorders suffer from substantial quality of life impairments and demonstrate reduced ability to participate in their professional and social life [2,3]. Depression is the most prevalent mental disorder in the Japanese population [4,5]. Insomnia is a predictor of subsequent mental illness, especially depression [6,7]. More than 85% of patients with depression are reported to have insomnia [8,9]; it remains the most common unresolved symptom of depression even after mood improvement by pharmacological treatment [10].

Examining the relationship between sleep and depression may be important for understanding the aetiology of affective disorders. Most studies that use electroencephalography (EEG) to objectively assess sleep have been conducted using polysomnography in the laboratory [11]. The use of the laboratory polysomnogram as the gold standard has limited validity because subjects tend to receive worse quality of sleep in the laboratory than at home [12]. Here, we used 1-channel (1-Ch) sleep EEG measurements at home to analyse the participant’s sleep. This 1-Ch EEG was reported to have acceptable reliability with simultaneously recorded polysomnography with an epoch-to-epoch agreement of 86.9% [13] and kappa of 0.64–0.75 [13,14]. Home monitoring sleep EEG, which can be used by many participants, may provide epidemiological evidence to support previous studies and provide new insights.

Although various studies have been conducted on EEG indices as biomarkers of depression, they are still in the process of being established and additional epidemiological studies are required. To the best of our knowledge, this is the first epidemiological study using home 1-Ch EEG to analyse the association between sleep and depressive symptoms. This study aimed to identify the EEG parameters associated with depressive symptoms based on the results of a questionnaire survey and EEG measurements in a large population.

## 2. Materials and Methods

### 2.1. Participants

A cross-sectional study was conducted as part of the Night in Japan Home Sleep Monitoring (NinJaSleep) sleep and mental health epidemiological study [15]. Participants included government employees of Koka City, which is a rural city in the Shiga Prefecture of Japan. Among the 1669 employees, 89 employees were excluded because of long-term leaves (more than 30 days), including sick, maternity, and childcare leaves. A total of 1580 questionnaires were distributed on 3 September 2018. Of these, 1479 employees returned the questionnaire (93.6%: 1479/1580). Exclusion criteria were usage of antidepressants (*n* = 33), having severe sleep apnoea with a respiratory event index (REI) ≥30 (*n* = 26), and missing items in the questionnaire (*n* = 30). Thus, 1390 participants were eligible for analysis. Of these, 650 underwent sleep monitoring with 1-Ch EEG at home (participants) and the others did not perform home EEG recordings (non-participants, *n* = 740) (Figure 1).

The Ethics Committee of Shiga University of Medical Science approved the study protocol (R2017-111). The study was registered at UMIN-CTR (UMIN000028675, registered on 15 August 2017) and ClinicalTrials.gov (NCT03276585, registered on 3 August 2017). Informed consent was obtained from each participant prior to participation. The datasets analysed in the current study are available from the corresponding author upon reasonable request.

The results of this survey were returned to the participants, including questionnaires and sleep monitoring data. The results were summarised into posters and displayed in the cafeteria of the city hall. Periodically, workshops were held to explain the results. We also provided opportunities for participants to consult physicians/researchers individually after the workshops.

### 2.2. Questionnaires

The Patient Health Questionnaire-9 (PHQ-9) is a 9-item questionnaire designed to screen for depression/depressive symptoms in clinical and research settings [16]. The standard cut-off score for screening to identify possible major depression/depressive symptoms is 10 or above. According to a meta-analysis, PHQ-9 had sensitivity and specificity of 0.85 (95% confidence interval (CI): 0.79–0.89) and 0.85 (95% CI: 0.82–0.87), respectively [17]. The PHQ-9 contains items derived from the DSM-IV classification system pertaining to (1) anhedonia, (2) depressed mood, (3) trouble sleeping, (4) feeling tired, (5) change in appetite, (6) guilt or worthlessness, (7) trouble concentrating, (8) feeling slowed down or restless, and (9) suicidal thoughts; each of the 9 DSM-IV criteria are scored from “0” (not at all) to “3” (nearly every day) [16]. In previous studies, participants with a PHQ-9 ≥ 10 were classified as having depression [15,16], and this same cut-off was used in this study.

Bedtime, awakening time, and subjective sleep latency were asked in the questionnaire. Subjective sleep duration was calculated by subtracting subjective sleep latency from the time spent in bed (calculated from bedtime and awakening).

The occupation of each participant was classified into management and clerical occupations, educational instruction occupations, healthcare and healthcare support occupations, other occupations, and temporary and commissioned occupations.

### 2.3. Sleep Monitoring

We used SleepScope (SleepWell, Co., Ltd., Osaka, Japan), a single-channel portable EEG device, to monitor sleep EEG at home. Both the method and analysis of the SleepScope recordings are described in detail elsewhere [13,14]. Briefly, one electrode was placed in the middle of the forehead and the other electrode on the left mastoid. In addition, the data obtained by SleepScope were forwarded to cloud services (SEAS-G, SleepWell Co. Ltd., Osaka, Japan), in which spectral analysis of the EEG data was performed for every 30-s epoch; the data were classified into five sleep stages: wake, rapid eye movement (REM), stage N1, stage N2, and stage N3. This service was approved by the Japanese Medical Device Certification (225ADBZX00020000). The home sleep monitoring study was performed between 16 February 2018 and 20 September 2019, and all the participants in the home sleep monitoring study participated in the questionnaire survey.

To exclude severe sleep apnoea, which is associated with the fragmentation of sleep due to the repeated occurrence of end-apnoeic arousal throughout the night, we performed a home sleep apnoea test (PulSleep LS-140, Fukuda Denshi Co., Ltd., Tokyo, Japan). REI was calculated as the total number of respiratory events (apnoeas and hypopneas) divided by monitoring time [18]. Apnoea (cessation of breathing for at least 10 s) and hypopnea (more than 30% reduction in the amplitude of nasal pressure or respiratory effort associated with a >3% reduction in oxyhaemoglobin saturation for at least 10 s) [18] were manually scored by a certified polysomnographic technologist by the Japanese Sleep Research Society.

### 2.4. Statistical Analyses

Statistical analysis was performed using IBM SPSS Statistics (version 25.0; IBM, Armonk, NY, USA). Values are reported as the mean ± standard deviation (SD). A two-tailed test was used to determine the statistical significance. One-way analysis of variance was performed to analyse differences in PHQ-9 scores according to categories. Differences in sleep EEG variables between the groups with and without depressive symptoms were analysed using the t-test. Odds ratios (ORs) and the 95% CIs for depressive symptoms of the sleep EEG variables were calculated using logistic regression analysis. Multivariate logistic regression was used to control for potential confounders such as age, sex, body mass index (BMI), occupation, sleep period time (SPT), sleep latency (SL), REML (REM latency), awakening time, δ-power value of the first sleep stage per minute, and sleep efficiency (SE). Statistical significance was set at *p* < 0.05. Power analysis was performed by calculating the mean and standard deviation of both depressive and non-depressive groups and calculating the sample size required to achieve a prescribed statistical power (α = 0.05, β = 0.8) using the Power and Sample Size software ver. A57e8c3 (Department of Biostatistics, Vanderbilt University, Nashville, TN, USA; https://vbiostatps.app.vumc.org/ps/, accessed on 1 December 2021).

## 3. Results

Among the eligible participants, 19.4% (95/490) of males and 22.4% (202/900) of females had depressive symptoms (PHQ-9 ≥ 10) (*p* = 0.184). There were no significant differences in PHQ-9 scores between males and females (4.11 ± 4.88 vs. 4.47 ± 439, *p* = 0.188). When compared in 10-year age groups, only those aged 60 years and above had lower PHQ-9 scores (Appendix A).

Compared to the non-participants, the participants had significantly higher percentage of males, age, and PHQ-9 scores but significantly shorter subjective sleep duration (Table 1).

There was no significant difference between subjective sleep duration and objective SPT (362 ± 70.6 min vs. 364 ± 79.1 min, *p* = 0.527). The subjective SL was significantly longer than objective SL (20.8 ± 18.0 min vs. 18.0 ± 15.4 min, *p* < 0.001). The mean differences in sleep duration (2.19 ± 88.24 min) and SL (2.75 ± 21.9 min) were both less than 3 min.

Among our EEG-measured participants, 20.6% (134/650) had depressive symptoms (PHQ-9 ≥ 10) (Table 2). There were no significant differences between the two groups in age, sex, BMI, SPT, number of awakenings, δ-power per minute in the first sleep cycle, or SE (Table 2). Significantly longer SL (*p* = 0.024) and significantly shorter REML (*p* = 0.027) were observed among participants with depressive symptoms. Subjective sleep duration was significantly shorter than objective SPT among participants with depressive symptoms (355 ± 78.3 min vs. 369 ± 79.9 min, *p* = 0.006). There were no significant differences in PHQ-9 scores according to SPT (Appendix A).

Logistic regression analysis was performed for depressive symptoms according to the PHQ-9 score and EEG variables (Table 3). There was an association between SL and depressive symptoms (OR, 1.02; 95% CI: 1.01–1.04). REML was also associated with depressive symptoms (OR, 0.99; 95% CI: 0.99–1.00). The associations between BMI and depressive symptoms, sleep efficiency, and depressive symptoms showed similar trends, although the differences were not significant. According to the power analysis, sample sizes of 3145 and 183,153 were required to show significant differences in BMI and SE, respectively. Depressive symptoms were not associated with sex, occupational categories, SPT, SL, or δ-power per minute during the first sleep cycle.

## 4. Discussion

Depressive symptoms were associated with prolonged SL and reduced REML in a Japanese working population. There are only a few studies on EEG measurements in non-clinical settings [11].

A total of 650 (46.8%) city government employees participated in this home sleep monitoring study. The participants had a significantly higher PHQ-9 score and a significantly shorter subjective sleep duration compared with the non-participants. However, these differences may not have any significant effects. The mean PHQ-9 scores were less than 5 points in both groups, and PHQ-9 scores ≥ 10 were classified as having depression [16,17]. The prevalence of depressive symptoms (PHQ-9 ≥ 10) was similar between participants and non-participants. Subjective sleep durations in both groups were shorter than the Organisation for Economic Co-operation and Development reported sleep duration in Japan in 2020 (442 min) [19].

Subjective-objective sleep discrepancies have been reported [20,21]. In this study, we did not find a significant discrepancy in subjective-objective sleep duration. Subjective SL was significantly longer than objective SL. However, the mean difference in SL was only less than three minutes. Previous studies reported that people with depressive symptoms had a discrepancy in subjective-objective sleep duration [21]. Similarly, in our study, participants with depressive symptoms had a significant discrepancy in subjective-objective sleep duration. Our exclusion of depressed subjects may have reduced the subjective-objective sleep discrepancies.

Both short and long sleep durations were reported to have an increased risk of depression [22]. In this study, we could not find a significant association between objective sleep duration and depressive symptoms (Table 2 and Appendix A). Some studies also found no association between sleep duration and depression [23,24]. This discrepancy might be due to differences in the methods used to assess sleep duration, potential confounding factors, definitions of short/long sleep, and the age of participants [22].

Since the 1960s, sleep studies using polysomnography have shown that depression is associated with typical changes in sleep EEG structure [10,11,12]. Objective sleep changes in patients with depression comprise impaired sleep continuity (prolonged SL, increased number of intermittent awakenings, early morning awakening), reduced slow-wave sleep, and changes in REM sleep (shortened REML, prolonged first REM period, and elevated REM density particularly during the first REM period) [9,11,25].

A strong association was reported between REM sleep dysregulation and the course of major depressive disorder [26]. We found that depressive symptoms were associated with short REM latency. Changes in REM sleep often persist beyond actual depressive episodes. This increases vulnerability to recurrence and relapse, which may negatively affect treatment efficacy in general [27,28]. Antidepressants affect sleep architecture and generally tend to improve sleep disturbances in depression. Most antidepressants suppress REM sleep, thus delaying the onset of REM sleep and reducing its amount [25,29,30,31]. Our exclusion of antidepressant users from this study may contribute to the present difference in REML between subjects with and without depressive symptoms.

Different antidepressants are developed to positively modulate the serotoninergic (5-HT) and noradrenergic (NE) systems in the central nervous system [32]. During REM sleep, the REM-off neurons stop firing or significantly decrease the firing rate. Almost all the REM-off neurons are monoaminergic in the locus coeruleus as NE neurons, in the dorsal raphe as the serotoninergic (5-HT) neurons, and in the tuberomammillary nucleus within the hypothalamic region as the histaminergic neurons [32]. Antidepressants that increase the level of 5-HT and NE or the affinity of their receptors in synapses strongly influence REM sleep, while those with no direct effects on the monoaminergic system do not suppress REM sleep or decrease REM sleep moderately [32].

SL was prolonged in subjects with depressive symptoms in our study population. A meta-analysis of actigraphy reported that depressive patients had slightly longer SL compared with healthy controls, but this difference was not statistically significant (standardised mean difference: −0.307, 95% CI: −0.648 to 0.035, *p* = 0.078) [33]. This difference may be due to differences in sleep monitoring. The 1-Ch EEG is superior to actigraphy in detecting sleep-wake detection and SL measurements [14].

Slow-wave sleep has been reported to be decreased in patients with depression [9,11]. Slow-wave sleep is mainly observed in the first cycle of sleep. Thus, we analysed the amount of slow-wave sleep (δ-power value) during the first sleep cycle. However, we did not detect a difference in the δ-power value of the first sleep cycle between participants with and without depressive symptoms (Table 2).

It is well-established that depression is more prevalent in females [34], and 58.8% of participants in this study were females. However, we could not find significant differences in the prevalence of depressive symptoms and PHQ-9 scores between males and females in our study population. Depressive symptoms were not associated with sex in the logistic regression analysis among participants with EEG measurements. Gender difference in depression may be negligible in this study.

There are some strengths to our study. The EEG measurements were conducted in an environment that reflects the conditions of daily life. In general, actigraphy and full-PSG are used to objectively measure sleep. However, full-PSG was considered unsuitable for large-scale epidemiological studies and repeated measurements because of the cost and effort involved in its implementation. For this reason, studies to objectively evaluate sleep are limited, owing to problems such as a small sample size. In addition, full-PSG is generally performed in a sleep laboratory, but the quality of the test is not as good as that of normal sleep at home [12]. In this study, we measured daily sleep using a simple 1-Ch EEG, and it is a pioneering effort to analyse depressive symptoms in a working population and not in clinical settings. Second, this study is not a cross-sectional study but a cohort study with annual follow-ups; thus, causality can be analysed in the future. We plan to investigate outcomes such as onsets and incidents of diseases and events. Third, since this study targets local government employees, the characteristics may be similar to those of the residents there, and a stable participation rate is expected in future follow-ups. Fourth, we performed sleep apnoea screening simultaneously. The importance of controlling sleep apnoea in epidemiological sleep studies has been underscored by recent studies [35,36]. We excluded subjects with severe sleep apnoea because apnoeic events cause sleep fragmentation and have deleterious effects on sleep quality. This exclusion enabled us to perform sleep assessments of higher quality.

This study has some limitations. The accuracy of our sleep analysis with 1-Ch EEG may be inevitably and slightly inferior to that of full-PSG. The 1-Ch EEG we used had acceptable reliability with PSG with 86.89% epoch-to-epoch agreement [13], and the kappa value was 0.64–0.75 [13,14]. Our target population of this study was limited to one region of Japan. In addition, the participation rate for EEG measurements was less than 50%, which should be improved in future follow-up studies. Sociodemographic and socioeconomic data were limited in this study. We plan to obtain more sociodemographic data in the next annual questionnaire survey in this population to analyse the effects of social support and social isolation on depression.

## 5. Conclusions

In conclusion, we found that prolonged SL and short REML were associated with depressive symptoms in a Japanese working population. We excluded subjects under antidepressant treatment and those with severe sleep apnoea, which may have contributed to the differences in sleep parameters. Our results suggest that the association between depressive symptoms and sleep latency was confirmed in a non-clinical epidemiological population. The 1-Ch EEG may be a useful tool to monitor sleep and screen depression/depressive symptoms in non-clinical settings. The mean difference between the subjective and objective assessments of sleep duration and latency was less than 3 min, indicating the accuracy of subjective assessments of these parameters in a non-clinical setting.

## Figures and Tables

**Figure 1 ijerph-19-02112-f001:**
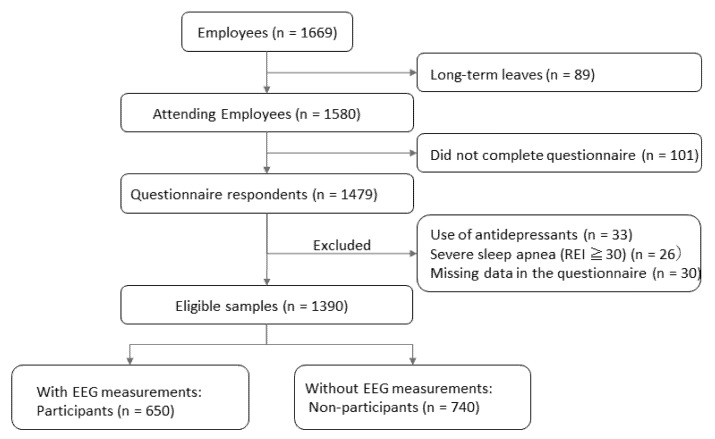
Flow diagram of the participants. REI: respiratory event index; EEG: electroencephalography.

**Table 1 ijerph-19-02112-t001:** Characteristics of participants and non-participants.

Variable	Participants	Non-Participants	*p*-Value
	(*n* = 650)	(*n* = 740)	
Male, *n* (%)	268 (41.2)	222 (30.0)	<0.001
Age, yrs. (SD)	44.7 (11.5)	46.6 (12.9)	0.003
BMI, kg/m^2^ (SD)	22.3 (3.32)	22.5 (3.70)	0.900
PHQ-9 score (SD)	4.79 (4.77)	3.90 (4.33)	<0.001
Depressive symptoms, *n* (%)	134 (20.6)	163 (22.0)	0.522
Subjective sleep duration, min (SD)	362 (70.6)	382 (74.3)	<0.001
SPT, min (SD)	364 (79.1)	-	
SL, min (SD)	18.0 (15.4)	-	
REML, min (SD)	78.2 (38.6)	-	
SE, %	87.0 (6.80)	-	
TST, min (SD)	335 (72.6)	-	
WASO, min (SD)	28.7 (22.6)	-	

Depressive symptoms, PHQ-9 ≥ 10; SD, standard deviation; BMI, body mass index; SPT, sleep period time; SL, sleep latency; REML, REM latency; SE, sleep efficiency; TST, total sleep time; WASO, wake time after onset; PHQ-9, Patient Health Questionnaire-9; SPT, SL, REML, SE, TST, and WASO were analysed from 1-Ch electroencephalography.

**Table 2 ijerph-19-02112-t002:** Comparison of sleep EEG variables in participants with and without depressive symptoms.

	No Depressive Symptoms (PHQ-9 < 10)	Depressive Symptoms (PHQ-9 ≥ 10)	*p*-Value
	(*n* = 516)	(*n* = 134)	
Age, years, mean (SD)	44.8 (11.5)	42.2 (11.5)	0.575
Male, *n* (%)	221 (42.8)	47 (35.1)	0.063
BMI, kg/m^2^, mean (SD)	22.3 (3.32)	22.7 (3.31)	0.205
SPT, min (SD)	362 (78.9)	369 (79.9)	0.360
SL, min (SD)	17.3 (13.8)	20.7 (20.3)	0.024
REML, min (SD)	79.9 (39.5)	71.7 (34.3)	0.027
Awakening, times (SD)	24.8 (12.1)	25.6 (11.6)	0.451
δ-power value of the first sleep cycle, μV^2^/min (SD)	2220 (1700)	2050 (1330)	0.284
SE, %	87.0 (6.74)	87.1 (7.06)	0.871

EEG, electroencephalography; SD, standard deviation; BMI, body mass index; SPT, sleep period time; SL, sleep latency; REML, REM latency; SE, sleep efficiency; PHQ-9, Patient Health Questionnaire-9.

**Table 3 ijerph-19-02112-t003:** Sleep 1-Ch EEG variables as a risk for depressive symptoms.

Model Terms	OR	95% CI	*p*-Value
Age			
1-year increment	0.98	0.96–1.00	0.108
BMI			
1-unit increment	1.06	0.99–1.12	0.056
Sex			
Female	1.38	0.85–2.24	0.187
Occupation			
Management and clerical	1.00	Reference	
Educational instruction	0.89	0.52–1.51	0.656
Healthcare and healthcare support	0.90	0.45–1.78	0.755
Temporary and commissioned	1.92	0.73–5.01	0.187
Others	1.53	0.53–4.39	0.428
SPT			
1-min increment	1.00	0.99–1.00	0.735
SL			
1-min increment	1.02	1.01–1.04	0.009
REML			
1-min increment	0.99	0.99–1.00	0.040
Awakening			
1-time increment	1.02	0.99–1.04	0.127
δ-power value of the first sleep stage per minute			
1 increment	1.00	1.00–1.00	0.114
SE			
1-percent increment	1.05	0.99–1.11	0.056

EEG, electroencephalography; OR, odds ratio; CI, confidence interval; bold values show statistical significance; SD, standard deviation; BMI, body mass index; SPT, sleep period time; SL, sleep latency; REML, REM latency; SE, sleep efficiency.

## Data Availability

The datasets analysed in the current study are available from the corresponding author upon reasonable request.

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
