# Peer review of "Prolonged Sleep Latency and Reduced REM Latency Are Associated with Depressive Symptoms in a Japanese Working Population"

_ijerph, 2022, doi:10.3390/ijerph19042112_

Round 1

Reviewer 1 Report

The manuscript presented for review describes the relationship between depression symptoms and prolonged sleep latency and REM.
The results are predictable; one of the symptoms of depression is sleep disturbance, including difficulty falling asleep.

Similar studies are available, but the value of this study is the large number of participants and a simpler, objective method of testing sleep than PSG, which may be used in the patient's home.   - line 36 - insomnia as a predictor of depression? The vast majority of insomnia is caused by physical and mental illnesses, including depression - the questionnaire used in the study diagnoses the presence of depressive symptoms rather than the depression itself. Many mental disorders can develop such symptoms, but depression is not e.g. schizophrenia, drug abuse, adjustment disorders. This should be included in the title and throughout the text. - the beginning of the results section duplicates the information contained in the methodology and also duplicates the data provided in the tables - precise data should not be given in the discussion, they are included in the results section and in the tables - line 194 - this information is already included in the introduction
- line 194-201 depression is not diagnosed only on the basis of symptoms, but also to the exclusion of diseases with similar symptomatology. This section of the discussion is redundant and is not related to the purpose of this study - the discussion is of poor quality - poor in relation to the main topic, a large part does not relate to the aim of the study, its structure is incorrect - line 234 - repeating the word "strengths"

Author Response

Reviewer1

  • The manuscript presented for review describes the relationship between depression symptoms and prolonged sleep latency and REM.
    The results are predictable; one of the symptoms of depression is sleep disturbance, including difficulty falling asleep. Similar studies are available, but the value of this study is the large number of participants and a simpler, objective method of testing sleep than PSG, which may be used in the patient's home.  

Response

Thank you very much for your comments. We have responded to all your comments and revised the manuscript accordingly.

  • - line 36 - insomnia as a predictor of depression? The vast majority of insomnia is caused by physical and mental illnesses, including depression - the questionnaire used in the study diagnoses the presence of depressive symptoms rather than the depression itself. Many mental disorders can develop such symptoms, but depression is not e.g. schizophrenia, drug abuse, adjustment disorders. This should be included in the title and throughout the text.

Response

Thank you very much for your comment. We changed from “depression” to “depressive symptoms” throughout the manuscript, including the title, to describe our participants with PHQ-9≥10. This change may help avoid confusion between depressive disorder and depressive symptoms. We agree that many mental disorders can develop some symptoms of depression.

  • - the beginning of the results section duplicates the information contained in the methodology and also duplicates the data provided in the tables- precise data should not be given in the discussion, they are included in the results section and in the tables 

Response

We deleted the first sentence of the results section. The data that was already presented in the tables were deleted in the text of the results section. In addition, the data in the discussion were deleted or moved to the results section.

  • - line 194 - this information is already included in the introduction

Response

We deleted this sentence.

  • - line 194-201 depression is not diagnosed only on the basis of symptoms, but also to the exclusion of diseases with similar symptomatology. This section of the discussion is redundant and is not related to the purpose of this study- the discussion is of poor quality - poor in relation to the main topic, a large part does not relate to the aim of the study, its structure is incorrect 

Response

We deleted lines 194–201. We also revised the discussion section to make it more meaningful. Thank you.

- line 234 - repeating the word "strengths"

Response

We deleted the word “strengths” and revised this sentence.

Reviewer 2 Report

In the present report, Authors used an at home evaluation of sleep quality in a population of depressed patients. The work is very inetersting, My only concers is related to sex-aggregated data. It is well known that female populatin have higher incidence of depression, indeed the population in the study is male onle around 41%. I suggest to present the disaggregated data per sex or at least to take into account in the discussion the possible gender effect.

Author Response

Reviewer 2

  • In the present report, Authors used an at home evaluation of sleep quality in a population of depressed patients. The work is very inetersting, My only concers is related to sex-aggregated data. It is well known that female populatin have higher incidence of depression, indeed the population in the study is male onle around 41%. I suggest to present the disaggregated data per sex or at least to take into account in the discussion the possible gender effect.

Response

Thank you very much for your comments. We have added data about sex and depression in the first paragraph of the results section. We have also added a paragraph on gender differences in the discussion section.

Reviewer 3 Report

Authors discussed about influence of depression in sleep outcome and this is an interesting topic even if it was already deeply explored in current literature.

An important issue emerge after reading this good article which is the place of social determinants in the development of depression related problems. The risk of depression is extremely dependent of socioeconomic status through multiple dimension of capital (human, social, etc...) and socioeconomic position. I strongly recommended authors to include a paragraph to extend/link their findings to current or popular approachs used in social epidemiology or sociology to assess such association. It is important because 1) social support of each participants depends on his social capital (social network, neighbourhood cohesion, etc..) and his parental socioeconomic status which affects his living conditions and his academic performance (due to necessity to work or not besides school, distance from home to university, etc...). This relationship was described recently and it was demonstrated objectively with quantitative measures such as actigraphy.

The data used by authors have for sure sociodemographic and socioeconomic data like education, neighborood, income, occupation, etc...which can support additional analysis. It will be interesting to see how current results will be distributed among participants in terms of objective sleep measurement. Authors may also provided a table of a graph showing difference between high educated individuals (with an university degree) and low educated individuals (not more than high school). Authors can also divided results by age group and sex to provide a more accurate profile of their current groups (with depression and without depression). Authors may used following articles to build their discussion:

1-Zhai L, Zhang H, Zhang D. SLEEP DURATION AND DEPRESSION AMONG ADULTS: A META-ANALYSIS OF PROSPECTIVE STUDIES. Depress Anxiety. 2015

2-Wang YQ, Li R, Zhang MQ, Zhang Z, Qu WM, Huang ZL. The Neurobiological Mechanisms and Treatments of REM Sleep Disturbances in Depression. Curr Neuropharmacol. 2015

3-Yu J, Rawtaer I, Fam J, Jiang MJ, Feng L, Kua EH, Mahendran R. Sleep correlates of depression and anxiety in an elderly Asian population. Psychogeriatrics. 2016

Author Response

Reviewer 3

  • Authors discussed about influence of depression in sleep outcome and this is an interesting topic even if it was already deeply explored in current literature.

    An important issue emerge after reading this good article which is the place of social determinants in the development of depression related problems. The risk of depression is extremely dependent of socioeconomic status through multiple dimension of capital (human, social, etc...) and socioeconomic position. I strongly recommended authors to include a paragraph to extend/link their findings to current or popular approachs used in social epidemiology or sociology to assess such association. It is important because 1) social support of each participants depends on his social capital (social network, neighbourhood cohesion, etc..) and his parental socioeconomic status which affects his living conditions and his academic performance (due to necessity to work or not besides school, distance from home to university, etc...). This relationship was described recently and it was demonstrated objectively with quantitative measures such as actigraphy.

Response

Thank you very much for your encouraging comments. We are very interested in the effects of social support, social isolation, and job stress on depression. Our current dataset did not contain sufficient sociodemographic and socioeconomic data. We have added a description of the future study plan in the limitations section in the discussion.

  • The data used by authors have for sure sociodemographic and socioeconomic data like education, neighborhood, income, occupation, etc...which can support additional analysis. It will be interesting to see how current results will be distributed among participants in terms of objective sleep measurement. Authors may also provided a table of a graph showing difference between high educated individuals (with an university degree) and low educated individuals (not more than high school). Authors can also divided results by age group and sex to provide a more accurate profile of their current groups (with depression and without depression).

Response

We have added a paragraph in the methods section on how we distributed the results to the participants.

We have not gathered sufficient information on sociodemographic and socioeconomic data yet (such as education, neighborhood, and income). We will collect these data in the annual questionnaire survey in this population in the future. This manuscript focused on sleep EEG and depression/depressive symptoms. We would like to present detailed socio-demographic data analysis in a future study.

We have added descriptions of the association between depression and age/sex in the discussion section.

  • Authors may used following articles to build their discussion:
    1-Zhai L, Zhang H, Zhang D. SLEEP DURATION AND DEPRESSION AMONG ADULTS: A META-ANALYSIS OF PROSPECTIVE STUDIES. Depress Anxiety. 2015

2-Wang YQ, Li R, Zhang MQ, Zhang Z, Qu WM, Huang ZL. The Neurobiological Mechanisms and Treatments of REM Sleep Disturbances in Depression. Curr Neuropharmacol. 2015

3-Yu J, Rawtaer I, Fam J, Jiang MJ, Feng L, Kua EH, Mahendran R. Sleep correlates of depression and anxiety in an elderly Asian population. Psychogeriatrics. 2016

Response

We have revised the discussion section using the references you suggested. Thank you.

Round 2

Reviewer 3 Report

Good improvement of the initial version performed by authors.

Manuscript is in good shape.